

# A demographic history of a population of howler monkeys (*Alouatta palliata*) living in a fragmented landscape in Mexico

Jurgi Cristóbal Azkarate[1], Jacob C. Dunn[1,2], Cristina Domingo Balcells[3] and Joaquim Veà Baró[4,†]

[1] Division of Biological Anthropology, University of Cambridge, Cambridge, United Kingdom
[2] Animal and Environment Research Group, Anglia Ruskin University, Cambridge, United Kingdom
[3] Instituto de Biología, Universidad Nacional Autónoma de México, Mexico City, Mexico
[4] Centre Especial de Recerca en Primats, Facultat de Psicologia, Universitat de Barcelona, Barcelona, Spain
[†] Deceased.

## ABSTRACT

Long-term field studies are critical for our understanding of animal life history and the processes driving changes in demography. Here, we present long-term demographic data for the northernmost population of mantled howler monkeys (*Alouatta palliata*) residing in a highly anthropogenically fragmented landscape in Los Tuxtlas, Mexico. We carried out 454 monthly group visits to 10 groups of mantled howler monkeys between 2000 and 2011. The population remained relatively stable over the 11-year study period, with an overall increase in the total number of individuals. Birth rates and inter-birth intervals were comparable to those of howler monkeys at non-fragmented sites, suggesting that living in a fragmented landscape did not affect the reproductive output of our study population. Moreover, despite the landscape, dispersal events were commonplace, including many secondary dispersals (individuals emigrating from groups that they had previously immigrated into). Finally, we found a marked effect of seasonality on the dynamics of our study population. In particular, the period of lowest temperatures and resource scarcity between November and March was associated with higher mortality and reproductive inhibition, while the period of resource abundance between April and May was associated with the majority of conceptions and weaning of offspring. This, in turn, could be influencing dispersal patterns in our study area, as male howler monkeys seem to time some of their immigrations into new groups to coincide with the start of the period of higher fertility, while females preferentially joined new groups several months before the onset of this period. These data have important implications for the conservation and management of howler monkeys in fragmented landscapes, as well as for our understanding of the effect of seasonality over howler monkey dispersal, reproduction and survival.

## INTRODUCTION

Long-term field studies of primates, i.e., studies that cover at least an important proportion of individual life cycles, are critical for our understanding of life history and the processes

Corresponding authors
Jurgi Cristóbal Azkarate, jca40@cam.ac.uk
Jacob C. Dunn, jcd54@cam.ac.uk

driving changes in demography (*Kappeler & Watts, 2012*). However, field studies that have lasted long enough to provide data spanning several generations have only been carried out on a very small number of primate species (*Kappeler & Watts, 2012*), and the long-term studies that do exist are usually limited to one or a handful of sites across the species' distribution. Given that demographic patterns are contingent on local climate and vegetation, a comprehensive understanding of the factors determining dispersal processes, mortality and fertility of primates requires long-term studies to be conducted not only in different taxa, but also in different landscapes and locations. Such studies are particularly important in modified habitats, where monitoring demographic parameters in threatened populations may be critical for primate conservation.

Long-term data on howler monkey (*Alouatta* spp.) demography is limited to studies of red howler monkeys in Venezuela (*A. arctoidea*) (*Crockett & Rudran, 1987*; *Rudran & Fernandez-Duque, 2003*), mantled howler monkeys in Panama (*A. palliata aequatorialis*) (*Milton, 1982*; *Milton, 1990*; *Milton, 1996*) and Costa Rica (*A. p. palliata*) (*Glander, 1992*; *Clarke et al., 2002*; *Zucker & Clarke, 2003*; *Clarke & Glander, 2010*), black and gold howler monkeys (*A. caraya*) in Argentina (*Kowalewski & Zunino, 2004*; *Zunino et al., 2007*), and Central American black howler monkeys (*A. pigra*) in Mexico (*Dias et al., 2015*).

Here, we present eleven years of demographic data from ten groups of mantled howler monkeys (*Alouatta palliata mexicana*) residing in a highly fragmented landscape in the Los Tuxtlas Biosphere Reserve, Mexico. Despite howler monkeys having been studied since the 1980s in Los Tuxtlas (*Cristóbal-Azkarate & Dunn, 2013*), our knowledge of reproduction, mortality and migration in this subspecies is very limited, and what data are available are mostly based on indirect evidence from single population censuses and anecdotal observations (*Estrada & Coates-Estrada, 1996*; *Cristóbal-Azkarate, Dias & Veà, 2004*; *Cristóbal-Azkarate et al., 2005*).

The motivation for this study was twofold. Firstly, we wanted to analyse population size, dispersal patterns and reproductive parameters such as birth rates and inter-birth intervals in order to understand the consequences of living in anthropogenically fragmented landscapes in this taxon. Long-term data, from several groups, is essential in order to obtain reliable data on such measures, as variation might be expected across both years and groups, and reproductive parameters require observations over several consecutive years. Owing to widespread habitat loss and fragmentation throughout its range, the remaining population of *A. p. mexicana* is now restricted to highly fragmented forested areas which has led it to be listed as critically endangered by the IUCN (*Cuarón et al., 2008*). Information generated by this study will be useful to understand the capacity of these primates to adapt to transformed landscapes and help develop informed projections of the conservation risk of this subspecies.

Secondly, we wanted to analyse the relationship between seasonality and howler monkey dispersal patterns, reproduction and survival. Los Tuxtlas represents the northernmost limit of mantled howler monkey distribution, and is near the northern limit of the distribution of the genus (*Cortés-Ortiz, Rylands & Mittermeier, 2015*; *Rylands et al., 2006*). Previous studies indicate that winter is a period of energetic stress due to the combined effect of increased thermoregulatory demands and lower food availability (Cristóbal-Azkarate J,

Booth RK, Dunn JC, Vea JJ, Wasser SK, 2017, unpublished data; *Dunn et al., 2013*), but whether this has any impact over the fitness of this howler monkey population is yet to be studied. Establishing correlates between climate and life history parameters will allow us to better understand the challenges howler monkeys face at the extreme limits of their distribution, and the responses they develop to cope with them.

## METHODS

### Ethics statement

This study is based on observational data and there was no direct interaction with the study subjects. We were granted access to the study site by local communities, landowners, and the Los Tuxtlas Biosphere Reserve, part of the National Commission of Natural Protected Areas of Mexico (CONANP). All research adhered to the American Society of Primatologists Principles for the Ethical Treatment of Non-Human Primates and to the legal requirements of Mexico.

### Study species

Five subspecies of mantled howler monkeys (*Alouatta palliata*) are currently recognised: *A. p. mexicana*, *A. p. palliata* and *A. p. aequatorialis*, *A. p. coibensis*, and *A. p. trabeata*. These subspecies are distributed from south-east Mexico to northwest Peru (*Cortés-Ortiz, Rylands & Mittermeier, 2015*).

Mantled howler monkeys are seasonally folivorous, with leaves contributing over 80% of food intake when fruit is scarce (*Milton, 1980*; *Glander, 1981*; *Cristóbal-Azkarate & Arroyo-Rodríguez, 2007*; *Dunn, Cristóbal-Azkarate & Veà, 2010*). This degree of folivory has been associated with their small home range size compared to other more frugivorous species (*Milton & May, 1976*) and primates living in small home ranges are considered to be more resistant to habitat fragmentation (*Cowlishaw & Dunbar, 2000*).

Gestation lasts six months in mantled howler monkeys (*Glander, 1980*) and weaning occurs at approximately 18–20 months of age (*Carpenter, 1934*; *Clarke, 1990*; *Domingo-Balcells & Veà-Baró, 2009*). Age of first reproduction for females is approximately 41–43 months and males reach maturity at approximately 48 months of age (*Glander, 1980*; *Domingo-Balcells & Veà-Baró, 2009*). This species is characterized by bisexual emigration of juveniles; males typically emigrate at around 22 months of age and females typically emigrate at around 33 months of age (*Glander, 1992*). However, it has been suggested that in Los Tuxtlas juveniles may occasionally emigrate as early as 14 months of age (*Domingo-Balcells & Veà-Baró, 2009*). Accordingly, the 11-year duration of our study covers an important proportion of a howler monkey's life cycle. Recent evidence indicates that secondary dispersal (individuals emigrating from groups that they have previously immigrated into) also exists in mantled howler monkeys and that this can be driven by the sex ratio of groups (*Clarke & Glander, 2010*). Dispersal patterns can be disturbed by relatively low levels of fragmentation (*Chiarello & De Melo, 2001*), as howler monkeys are highly arboreal and spend almost all of their time in the upper canopy, very rarely coming to the ground (*Mendel, 1976*).

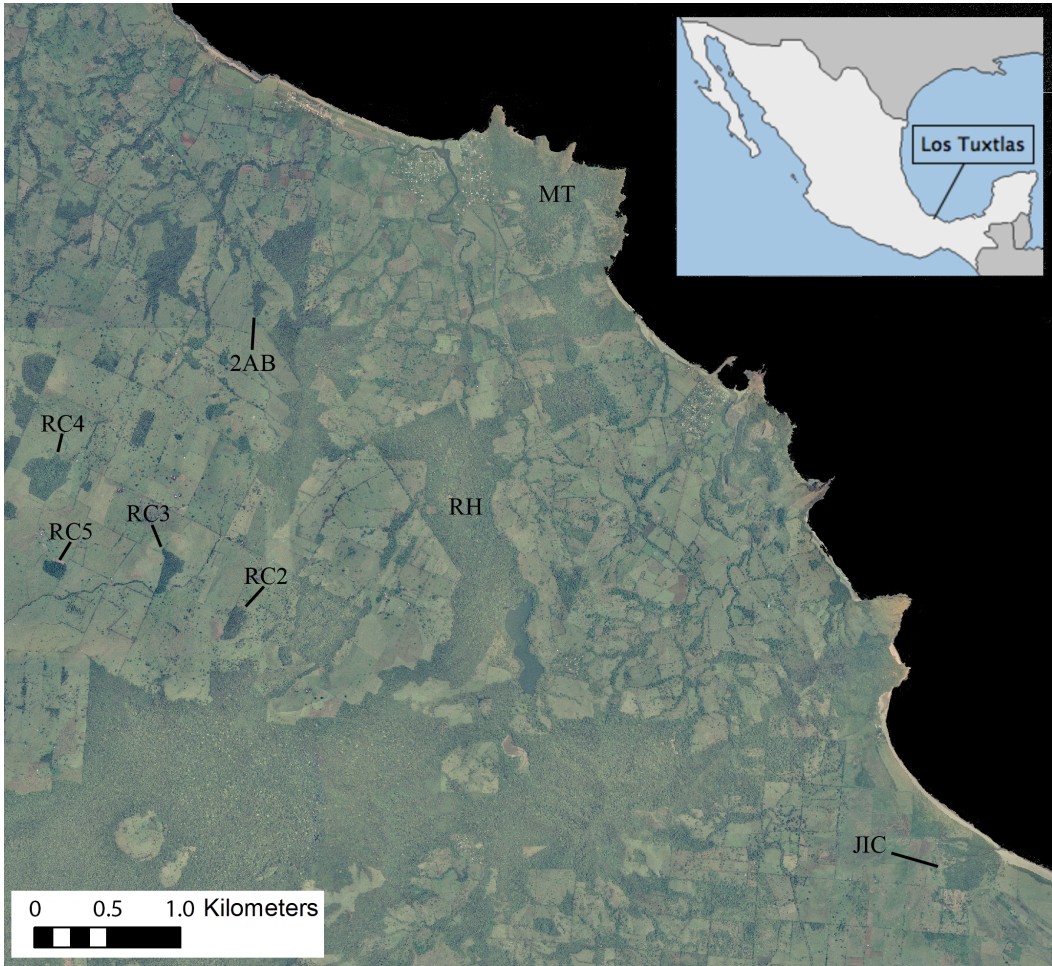

**Figure 1** **Ortophoto obtained from INEGI (http://www.inegi.org.mx) of our 7,500 ha study area in the Los Tuxtlas Biosphere Reserve, Veracruz, Mexico, indicating the forest fragments inhabited by the 10 study groups.** Areas in dark green represent forest, light green pasture and black the sea. Note that the RH fragment has recently connected to continuous forest through regrowth of secondary vegetation, but during the period that this group was studied there was no such connection.

## Study site

The Los Tuxtlas Biosphere Reserve represents the northernmost limit of tropical rainforest distribution in the Americas (*Guevara-Sada, Laborde & Sánchez-Ríos, 2004*). Our study site (18°39′21″–18°31′20″N and 95°9′14″–95°1′45″W; elevation 0–400 m a.s.l) covers approximately 7,500 hectares, and like many other regions throughout the tropics, it has suffered from extensive forest loss, transformation, and fragmentation, principally as a result of cattle farming (Fig. 1). This occurred predominantly between 1976–1986, and the great majority of the actual forest fragments were created during this time (*Cristóbal-Azkarate, 2004*). Nevertheless, compared to many other fragmented landscapes, it retains a relatively high level of connectivity, with live fences (i.e., several strands of barbed wire held up by a line of trees), riparian vegetation and isolated trees found between many fragments, and a mean distance to nearest fragment of 103 ± 172 m (*Arroyo-Rodríguez,*

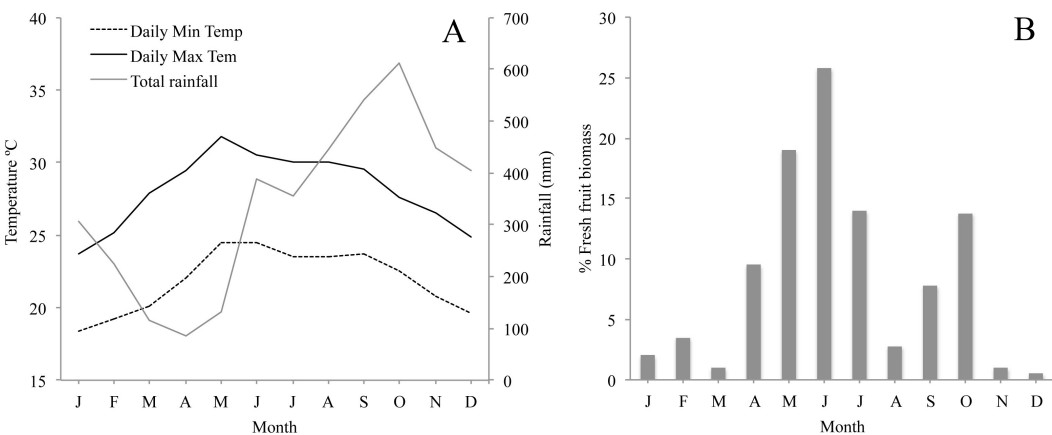

**Figure 2** (A) Monthly average minimum and maximum temperature and rainfall in the study area for the study period; and (B) Plant phenology in Los Tuxtlas adapted from *Dunn, Cristóbal-Azkarate & Veà (2010)* showing percentage of fresh fruit biomass each month.

*Mandujano & Benítez-Malvido, 2008*). It also contains areas of continuous forest in close proximity (less than 500 m) to the fragments (Fig. 1).

The climate in Los Tuxtlas is warm and humid with a mean annual temperature of 25 °C and rainfall of 4,900 mm (*Soto, 2004*). There is a dry season between March and May and a wet season from June to February. During the wet season there is also a period of strong winds and a considerable reduction in temperature between October and February (Fig. 2A). Long-term records of phenological data in the region show that there are two distinct peaks in fruit production: a primary peak at the end of the dry season—beginning of the rainy season (April–June), and a shorter, less intense secondary peak in the wet season (August–October), while fruit production abruptly falls to very low levels between November and March (Fig. 2B). The howler monkeys in Los Tuxtlas respond to the reduction in temperature and fruit availability between November and March by increasing their consumption of leaves and their foraging effort (*Dunn, Cristóbal-Azkarate & Veà, 2010*), which, in turn, has been associated to higher levels of physiological stress (*Dunn et al., 2013*). Therefore, we refer to this period as the "period of energetic stress".

## Study groups
We carried out the first census of our study site in 2000. Of the 55 forest fragments that are found in our study site, we found 21 to be inhabited with at least one howler monkey and recorded a population of 316 individuals living in 43 groups (*Cristóbal-Azkarate et al., 2005*). We began studying four of these groups intensively in 2000. Over the following 10 years, we studied six more groups as part of a programme of interdisciplinary research, for a total of 10 groups, which provided the data for our analyses (e.g., *Cristóbal-Azkarate et al., 2006*; *Cristóbal-Azkarate et al., 2007*; *Dunn, Cristóbal-Azkarate & Veà, 2010*; *Dunn et al., 2013*). Despite the wide-ranging nature of the research, we gathered basic demographic data, such as the number of individuals, age-sex composition, births, deaths and migrations, over the 11-year period.

## Data collection

We present demographic data from 10 groups of howler monkeys, representing 454 monthly group visits, which were carried out between 2000 and 2011 (Table 1). Given that the data has been pooled across several different studies, there is some discontinuity, with certain groups being studied for longer and/or more frequently than others (mean ± SD = 45.1 ± 29.7 monthly visits per group; Table 1). The study groups inhabited eight different forest fragments, which varied in size, shape and connectivity (Fig. 1).

We identified group members by the distinguishing colour patterns on their feet and tails, which are characteristic of this subspecies. We created an identity sheet for each individual as a reference in the field, drawing and making notes on the distinctive features. Each time we recorded a new individual in a group, we assigned it an age and sex using the classification system developed by *Domingo-Balcells & Veà-Baró (2009)*, which allows an age range to be estimated on the basis of morphological and behavioural characteristics.

## Demographic events

Throughout the study, we registered all demographic events in the groups, including: emigration, immigration, birth and death. However, given the low probability of observing these events directly, some of the events were also assumed to have occurred on the basis of changes in group composition and supporting evidence.

### Birth

We assumed a new individual had been born in a group when a new dependent infant, which was strongly associated with one of the group females, was observed in a group.

To calculate the mean annual birth rate for each group, we determined the number of births that had taken place per year for the mean number of adult females in the group. This allowed us to control for the effect of the number of females on birth rate. We defined the inter-birth interval (IBI) as the time that occurred between births for any given female.

### Emigration

We assumed an individual had emigrated from a group when all of the following criteria were met: (1) we had not observed the individual in the group for more than one month; (2) the last time we observed the individual it showed no sign of disease or injury; and (3) the last time we observed the individual it was fully weaned (unless emigrating with its mother). We also classified an individual as having emigrated if it was observed in a new group or as a solitary individual.

When an individual emigrated from the group it was born in, we defined this as 'natal emigration'. When an individual emigrated from a group that it had previously immigrated into, we defined this as 'secondary emigration'.

### Immigration

We assumed a new individual had immigrated into a group when, on first sighting, its estimated age was greater than the time passed since our last visit to the group (e.g., a new individual with an estimated age of 12 months was observed for the first time in a group, but the group was last visited 2 months ago).

Cristóbal Azkarate et al. (2017), *PeerJ*, DOI 10.7717/peerj.3547

**Table 1  Sampling chart of monthly visits to 10 groups of mantled howler monkeys in Los Tuxtlas, Mexico, between 2000 and 2011.** Months with at least one visit to a group are shown in dark grey. Detailed demographic data are also provided for all events for which we were able to determine the date with a maximum error of one month.

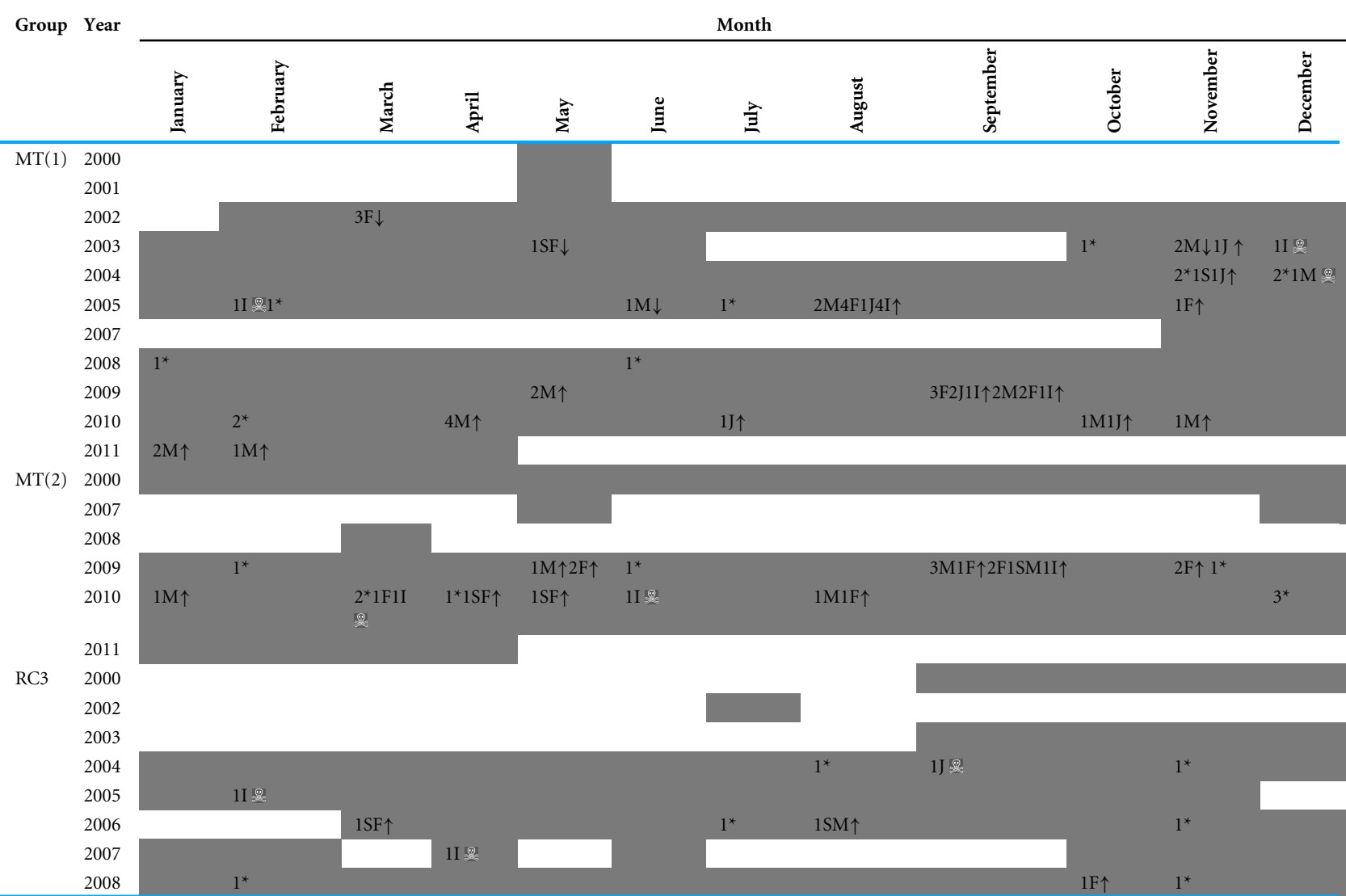

Cristóbal Azkarate et al. (2017), *PeerJ*, DOI 10.7717/peerj.3547

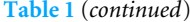

**Table 1** (*continued*)

| Group | Year | January | February | March | April | May | June | July | August | September | October | November | December |
|---|---|---|---|---|---|---|---|---|---|---|---|---|---|
| | 2009 | | | | | | | | | 1* | 1F↑ | 1SM↑ | 1F↑ |
| | 2010 | 1J↑ | 1I💀2M1J↑ | | | | | | 1M↑ | | 2* | | |
| | 2011 | | | | | | | | | | | | |
| 2AB | 2001 | | | | | | | | | | | | |
| | 2002 | | | | | | | | | | | | |
| | 2003 | | | | | | | | | | | | |
| | 2004 | 1M💀 1* | | | | | | | | | 1* | | 1J↑ |
| | 2005 | | | | | 1F↑ | | 1J↑ | 1* | | | | |
| | 2007 | | | | | | | | | | | | |
| | 2008 | | | | | | | | | | | | |
| | 2009 | | | | | | 2M1F1S↑ | | 1*1I↑ | | | | |
| | 2010 | | | | | | | | 1SM↑ | | 1* | 1SM↑ | |
| | 2011 | 1J↑ | | | | | | | | | | | |
| RH | 2001 | | | | | | | | | | | | |
| | 2002 | | | | | | | | | | | | |
| | 2003 | | | | | | | | | 1* | 1* | | 1I💀 |
| | 2004 | | 1J↑ | 1* | | | | | | | 1* | 1I💀 | 1* |
| | 2005 | | | 1* | | | | | | | | 1*1J↑ | |
| | 2006 | | | | | | 1* | 1F↑ | 1I💀 | | | | |
| | 2007 | | | | | | | | | | | | |
| | 2008 | | | | | | | | | | | | |
| JIC | 2001 | | | | | | | | | | | | |
| | 2002 | | | | | | | | | | | | |
| | 2004 | | | | | | | | | | | | |
| | 2005 | | | | | | | | | | | | |
| | 2007 | | | | | | | | | | | | |
| | 2008 | | | | | | | | | | | 1* | |
| | 2009 | | | | | | | | | | | | |

**Table 1** (*continued*)

| Group | Year | January | February | March | April | May | June | July | August | September | October | November | December |
|---|---|---|---|---|---|---|---|---|---|---|---|---|---|
| RC5 | 2007 | | | | | | | | | | | | |
| | 2008 | | | | | | | | | | | | |
| | 2009 | 1F1S1I↑  1* | | | | | | | | | | | 1F↑ |
| | 2010 | | | | | | | | | | 1* | 1I💀 | 1* |
| | 2011 | | | | | | | | | | | | |
| RC4(1) | 2000 | | | | | | | | | | | | |
| | 2001 | | | | | | | | | | | | |
| | 2004 | | | | | | | | | | | | 2M1F↑ |
| | 2005 | 1F↑ | | | | 1F↑1I💀 | | | | | | | |
| RC4(2) | 2004 | | | | | | | | | | | | |
| | 2005 | | | | | | | | | | | | |
| RC2 | 2000 | | | | | | | | | | | | |
| | 2001 | | | | | | | | | | | | |
| | 2004 | | | | | | | | | | | | 2F1I↑ 1M💀 |
| | 2005 | 1* | 1I💀 | | | | | | | 1F1SF↑ | | 2* | |

**Notes.**

*,  Birth; ↓, immigration; ↑, emigration;💀, death;  M,  Adult male;  F,  Adult female;  S,  Sub-adult (SM, Subadult male; SF, Subadult female);  J,  Juvenile];  I,  Infant.

### Death

We assumed an individual had died when at least one of the following criteria was met: (1) we found the body; (2) the individual went missing while still dependent on its mother's milk and its mother remained in the group; or (3) the individual went missing fully weaned, but was showing serious signs of injury or disease the last time it was observed.

### Disappeared

For some individuals it was not possible to determine with any confidence whether they had emigrated or died. Therefore, we recorded these individuals as disappeared.

## Statistical analyses

For the calculation of the seasonality of demographic events, IBI and birth rates, we considered only those events that were registered during periods in which the study groups were observed continuously and that could be assigned to a date with a maximum error of one month. In order to control for the effect that our slightly unbalanced sampling effort could have on the seasonality data, we weighted the original data by dividing the frequency of events per month by the number of different visits to the same group within a month (mean ± SD average visits per month = 37.6 ± 3.1, range = 35–42; Table 1). We used these weighted values to calculate the percentage of demographic events in each month.

We used ANOVAs to analyse the differences in annual birth rate and IBI among groups, and reported eta squared ($\eta^2$) as a measure of effect size (which is analogous to $R^2$ in regression analyses). Values of $\eta^2$ vary from 0 to 1 and values of 0.02, 0.13, and 0.26, and can be, as a rule of thumb, considered as small, medium and large effects, respectively (*Cohen, 1973*). We also used a Student's $T$ test to test the hypothesis that the death of a suckling offspring, ≤14 months of age (*Domingo-Balcells & Veà-Baró, 2009*), shortens the IBI by comparing the mean IBI of females with surviving and non-surviving offspring, and reported *Cohen (1977)* as a measure of effect size. For Cohen's d effect sizes of 0.2, 0.5, and 0.8, can be thought of as small, medium and large, respectively (*Cohen, 1977*).

To test for differences in the frequency of demographic events between the season of energetic stress (November–March, see above) and the rest of the year, as well as to test for statistically significant differences between peaks in demographic events at certain times of year compared to the rest of the year, we conducted Chi-squared ($X^2$) goodness of fit tests, with expected values being proportionally calculated according to the number of months used in the analysis. We calculated effect sizes for Chi-square tests using Cramer's phi coefficient ($\varphi$), whereby 0.1, 0.3, and 0.5 can be interpreted as small, medium and large effects (*Cramer, 1999*). Furthermore, in order to account for the underlying continuity of the time variable, we also used circular statistics to test for seasonality of demographic events (*Batschelet, 1981*). This approach has several advantages over those traditionally used by primatologists to test for seasonality (*Janson & Verdolin, 2005*; *Gogarten et al., 2014*). The mean vector length ($r$) obtained from circular statistics is well suited as an index of seasonality, as it provides a measure of how evenly events are distributed throughout the year. When events are spread evenly across months (not seasonal), $r$ is close to zero and when events are highly clustered at the same time of year (highly seasonal), $r$ is close to one.
**Table 2  Demographic data from 10 groups of mantled howler monkeys in Los Tuxtlas, Mexico, between 2000 and 2011.**

| Group | Fragment size (ha) | Study period | Adults start | Total start | Birth | Emigration | Immigration | Death | Disappeared | Adults end | Total end | Adult change | Total change |
|---|---|---|---|---|---|---|---|---|---|---|---|---|---|
| MT (1) | 63.8 | 2000–2011 | 2 | 2 | 16 | 23 | 24 | 3 | 0 | 11 | 16 | 9 | 14 |
| MT (2) | 63.8 | 2000–2011 | 13 | 18 | 13 | 12 | 12 | 3 | 10 | 11 | 18 | −2 | 0 |
| RC3 | 7.2 | 2000–2011 | 5 | 6 | 10 | 7 | 5 | 4 | 4 | 4 | 6 | −1 | 0 |
| 2AB | 3.6 | 2001–2011 | 5 | 5 | 8 | 9 | 4 | 1 | 2 | 3 | 5 | −2 | 0 |
| JIC | 6.9 | 2001–2011 | 2 | 2 | 3 | 0 | 4 | 0 | 0 | 7 | 9 | 5 | 7 |
| RH | 244 | 2001–2011 | 5 | 6 | 12 | 3 | 1 | 3 | 4 | 6 | 9 | 1 | 3 |
| RC5 | 5.9 | 2007–2011 | 3 | 4 | 3 | 3 | 1 | 1 | 0 | 3 | 4 | 0 | 0 |
| RC2 | 5.3 | 2004–2005 | 11 | 12 | 5 | 3 | 2 | 2 | 0 | 10 | 14 | −1 | 2 |
| RC4 (1) | 17.5 | 2004–2005 | 6 | 8 | 2 | 1 | 4 | 0 | 7 | 5 | 5 | −1 | −3 |
| RC4 (2) | 17.5 | 2004–2005 | 5 | 5 | 3 | 1 | 0 | 1 | 0 | 5 | 6 | 0 | 1 |
| **TOTAL** | | **2000–2011** | **57** | **68** | **75** | **62** | **57** | **18** | **27** | **65** | **92** | **8** | **24** |

We tested the statistical significance of the $r$ statistic using the Rayleigh test (*Batschelet, 1981*), which compares the data with the null hypothesis that demographic events have a random distribution across months. As we used monthly data for demographic events, rather than specific dates, we also used a correction factor ($c = 1.0115$) when calculating the $r$ statistic (*Batschelet, 1981*). To test for bimodal distribution in the data, we also calculated $r$ by doubling the angle calculated for each demographic event (*Batschelet, 1981*; *Janson & Verdolin, 2005*; *Gogarten et al., 2014*).

We carried out analyses in R 2.13.1 (*R Core Development Team, 2008*), testing for normality in the data and considering $p < 0.05$ as significant.

## RESULTS

Overall, we observed an increase in the number of individuals in our population between 2000 and 2011. Most of the study groups showed little change in the number of individuals and in the number of adult individuals from the start to the end of the eleven-year study period. However, two groups showed a substantial increase in number (Table 2). Migration was the principal cause of change in group size and composition, followed by births, then deaths. An overview of the demographic events for which we were able to determine the date with a maximum error of one month is given in Table 1.

### Births

We registered 75 births and at least two births were observed in all 10 of our groups (Tables 1 and 2). Of these, we were able to determine the date of birth to within one month on 49 occasions.

The mean birth rate per group was $0.42 \pm 0.32$ births per female per year ($N = 39$ births; Table 3). There were groups with no births in some years, while other groups had a birth rate as high as 1 in some years (indicating that all females of reproductive age gave birth

Cristóbal Azkarate et al. (2017), *PeerJ*, DOI 10.7717/peerj.3547

**Table 3** **Mean birth rate and inter-birth interval for 10 groups of mantled howler monkeys in Los Tuxtlas, Mexico, between 2000 and 2010, as well as other studies of howler monkeys in the Neotropics.** See methods for details of how these parameters were calculated.

| Study | Taxon | Group | Mean birth rate (births per female per year) | | | | Inter-birth interval (IBI) (months) | | | |
| | | | Mean ± SD | CI (95%) | N (years) | Range | Mean ± SD | CI (95%) | N (cases) | Range |
|---|---|---|---|---|---|---|---|---|---|---|
| Present Study | A. p. mexicana | MT (1) | 0.36 ± 0.26 | 0.08/0.63 | 6 | 0.00–0.80 | 11.0 ± 4.2 | −27.1/49.1 | 2 | 8–14 |
| Present Study | A. p. mexicana | MT (2) | 0.50 ± 0.25 | −1.74/2.74 | 2 | 0.32–0.68 | 20.6 ± 9.9 | 8.3/32.9 | 5 | 8–35 |
| Present Study | A. p. mexicana | RC3 | 0.56 ± 0.40 | 0.17/0.97 | 7 | 0.00–1.00 | 23.33 ± 13.5 | 9.1/37.5 | 6 | 13–50 |
| Present Study | A. p. mexicana | 2AB | 0.47 ± 0.32 | 0.13/0.81 | 6 | 0.00–1.00 | 39.5 ± 24.7 | −182.9/261.9 | 2 | 22–57 |
| Present Study | A. p. mexicana | JIC | 0.18 ± 0.24 | −0.2/0.55 | 4 | 0.00-0.50 | – | – | – | – |
| Present Study | A. p. mexicana | RH | 0.52 ± 0.17 | 0.25/0.79 | 4 | 0.33–0.75 | 15 ± 4.3 | 4.2/25.8 | 3 | 12–20 |
| Present Study | A. p. mexicana | RC5 | 0.25 ± 0.35 | −2.93/3.42 | 2 | 0.00–0.50 | – | – | – | – |
| Present Study | A. p. mexicana | RC2 | 0.35 ± 0.33 | −2.65/3.35 | 2 | 0.11–0.58 | – | – | – | – |
| Present Study | A. p. mexicana | RC4 (1) | 0.33 ± 0.58 | −1.1/1.77 | 3 | 0.00–1.00 | – | – | – | – |
| Present Study | A. p. mexicana | RC4 (2) | 0.33 ± 0.33 | −0.49/1.16 | 3 | 0.00–0.67 | – | – | – | – |
| **Total** | – | | **0.42 ± 0.32** | **0.3/0.51** | **39** | **0.00–1.00** | **21.6 ± 13.3** | **15.0/28.2** | **18** | **8–57** |
| *Cortés-Ortiz et al. (1994)* | A. p. mexicana | – | 0.62 | – | 5 | – | 15.79 | – | 19 | 10–21 |
| *Carrera-Sánchez, Medel-Palacios & Rodríguez-Luna (2003)* | A. p. mexicana | – | ≈0.5 | – | 14 | 0.25 –1.00 | 20.4 | – | 20 | 8–50 |
| *Arroyo-Rodríguez, Mandujano & Benítez-Malvido (2008)* | A. p. mexicana | – | – | – | – | – | 25.0 ± 3.0 | – | 4 | 23–29 |
| *Glander (1980)* | A. p. palliata | – | 0.22 | – | 7 | 0.07 –0.40 | 22.5 ± 0.6 | – | 16 | 18–25 |
| *Fedigan & Rose (1995)* | A. p. palliata | – | ≈0.5 | – | 8 | 0.00 - 1.00 | 19.9 | – | 24 | 9–40 |
| *Milton (1982)* | A. p. aequatorialis | – | – | – | – | – | 17 | – | 3 | – |
| *Crockett & Rudran (1987)* | A. arctoidea | – | 0.68 | – | 8 | 0.55 –0.88 | 17.4 ± 4.5 | – | 135 | 10–35 |
| *Strier, Mendes & Santos (2001)* | A. guariba | – | – | – | – | – | 22.8 ± 6.6 | – | 12 | 11–38 |
| *Rumiz (1990)* | A. caraya | – | 0.89 | – | 4 | – | 15.9 ± 3.7 | – | 30 | 12–22 |
| *Horwich et al. (2001)* | A. pigra | – | – | – | – | – | 19.4 | – | 64 | 10–35 |

in that year). Although mean birth rate ranged from $0.18 \pm 0.24$ to $0.56 \pm 0.40$ births per female per year across groups (Table 3), we found no statistically significant differences in mean birth rate among groups ($F_{9,29} = 0.57$, $p = 0.81$, $\eta^2 = 0.15$).

The mean IBI was $21.6 \pm 13.3$ months ($N = 18$; Table 3). Although mean IBI varied from $11.0 \pm 4.2$ months to $39.5 \pm 24.7$ months across groups (Table 3), we found no statistically significant difference among groups ($F_{4,13} = 1.68$, $p = 0.22$, $\eta^2 = 0.34$). We observed 12 IBIs in which the offspring from the first birth had survived until weaning (mean $= 26.1 \pm 14.1$ months) and six IBIs in which the offspring from the first birth had died before weaning (mean $= 12.7 \pm 4.5$ months), and found a significant reduction in IBI when the first offspring had died before weaning (mean difference $= 13.42$, 95% CI [3.89–22.95], $t = 3.01$, $df = 16$, $p = 0.009$, $d = 1.28$).

Although we observed births throughout the whole year, births were clearly seasonal. Seventy-four per cent of births occurred between October and March, with a main peak in November and a smaller second peak in February (Fig. 3A & Table 1). Accordingly, the bimodal $r$ statistic was highly significant (unimodal $r = 0.20$, $p = 0.14$; bimodal $r = 0.43$, $p < 0.001$). As the gestation time of howler monkeys is 6 months, and weaning occurs at approximately 18–20 months, these data indicate that conceptions leading to births and weaning peaked in May, after the period of energetic stress. In fact, the number of conceptions leading to births was not equally distributed, with fewer conceptions than expected during the period of energetic stress (10 conceptions between November and March) and more than expected during the rest of the year (39 conceptions between April and October) ($X^2 = 9.38$, $df = 1$, $p < 0.005$, $\varphi = 0.19$).

## Emigration

We recorded emigrations in all but one of our study groups, with 62 individuals emigrating from groups (Tables 1 and 2). Thirty-four of these were adults (13 adult males and 21 adult females), 7 were sub-adults (2 males, 2 females, and 3 of unknown sex), 13 were juveniles and 8 were infants.

We recorded 24 natal emigrations (52.8% of emigrations of known origin). One by an adult male, 6 by adult females, 3 by subadults, 11 by juveniles and 7 by infants. Of the 7 infants, 4 left the group soon after the birth of a sibling and 8 left the group soon after one or more individuals had immigrated into their group.

We recorded 22 secondary emigrations (47.8% of emigrations of known origin). Nineteen of these were adults (11 males and 8 females), 1 was a male sub-adult, 1 was a juvenile and 1 was an infant that transferred to the group together with her mother, and then left with her after one month of permanence in the group. The mean time that the individuals spent in a group prior to secondary transfer was $18.4 \pm 21.1$ months (range $= 1–90$ months).

On 16 occasions, we were not able to determine whether the emigrating individuals were born in the group they emigrated from or whether they had previously immigrated into the group. Accordingly these emigrations were of unknown origin and were not classified as natal or secondary.

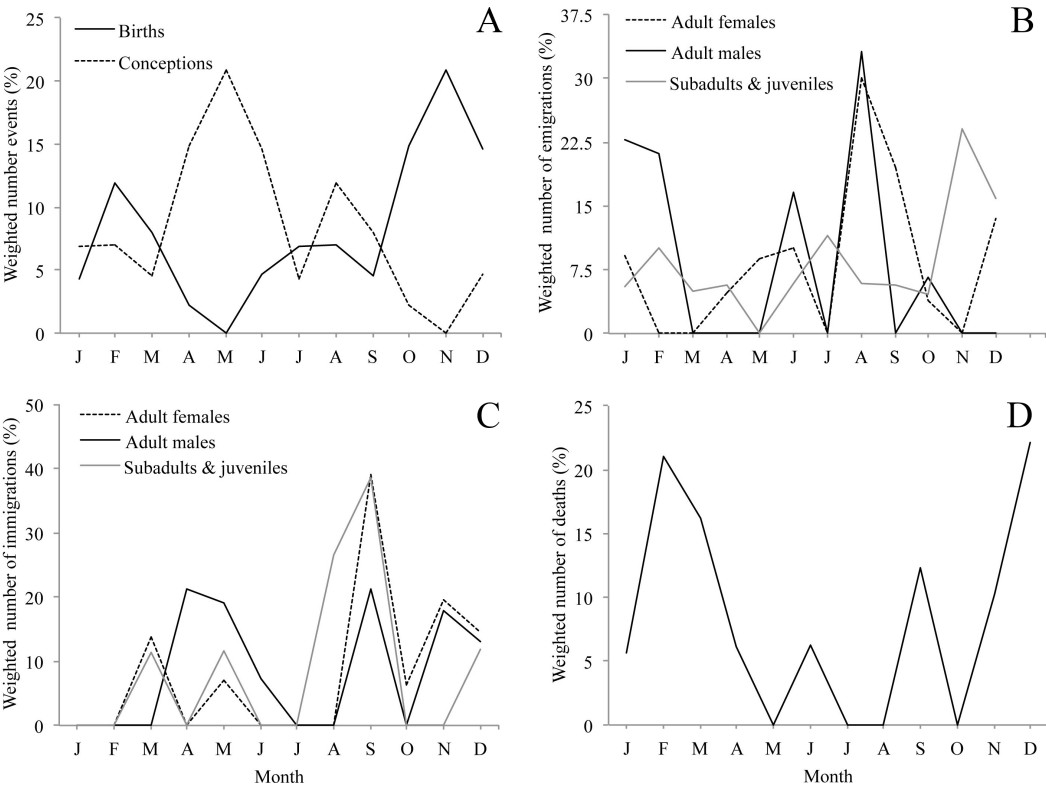

**Figure 3** **Seasonality of demographic events (A, birth, B, emigration, C, immigration, D, natural death, i.e., not associated with aggression) for 10 groups of mantled howler monkeys in the Los Tuxtlas Biosphere Reserve, Mexico.** Dispersal events of infants are not considered because they always occurred in the company of their mothers. Values are weighted by dividing the frequency of demographic events by the number of observations conducted each month.

Emigrations occurred throughout the year, but there were clear differences in emigration patterns among the age-sex classes (Fig. 3B, Tables 1 and 2). Male emigration peaked in August (33.1% of cases) and January–February (43.9%), when more emigrations occurred than expected ($X^2 = 8.57$, $df = 1$, $p < 0.005$, $\varphi = 0.66$; and $X^2 = 8.14$, $df = 1$, $p < 0.005$, $\varphi = 0.63$, respectively), female emigration showed a very clear peak in August–September (49.6% of cases), when more emigrations occurred than expected ($X^2 = 14.49$, $df = 1$, $p < 0.001$, $\varphi = 0.69$), and subadult and juvenile emigration showed a peak in November–December (38.9% of cases), when more emigrations happened than expected ($X^2 = 8.87$, $df = 1$, $p < 0.005$, $\varphi = 0.48$). However, the $r$ statistic for seasonality was non-significant in all cases, only trending towards significance in the females (unimodal $r = 0.34$, $p = 0.08$).

## Immigration

We recorded immigrations in all the forest fragments that we studied and in all but one of our study groups (this group was only followed for one year). We recorded 57 individuals immigrating into new groups (Tables 1 and 2); 46 were adults (22 males and 24 females), 6 were sub-adults (3 males and 3 females), 3 were juveniles and 2 were infants. Of these, we were able to determine the date of immigration to within one month in 41 cases (Table 1).

Immigration occurred throughout the year, but there were clear differences in immigration patterns among the age-sex classes (Fig. 3C, Tables 1 and 2). Adult male immigrations peaked in April–May (40.5% of cases), when more immigrations occurred than expected ($X^2 = 5.88$, $df = 1$, $p < 0.05$, $\varphi = 0.39$) and again in September–December (52.2%), though this was not significantly more than expected by chance. Female immigration showed a clear peak from September–December (79.3% of cases), when more immigrations occurred than expected ($X^2 = 12.89$, $df = 1$, $p < 0.001$, $\varphi = 0.92$), and subadults and juveniles showed a peak in immigration between August–September (65.3% of cases), when more immigrations occurred than expected ($X^2 = 12.10$, $df = 1$, $p < 0.001$, $\varphi = 1.51$). Despite these peaks in immigration, the $r$ statistic for seasonality was non-significant in all cases.

### Deaths

We registered 18 deaths, and at least one death was registered in eight of the 10 groups (Table 2). Thirteen of these individuals were infants: eight were younger than four months of age, three between four and eight months, and two were 10 months of age. Of these, we only observed one death directly, when, a one-month old infant died shortly after we found it lying by its dead mother which was seemingly killed by another howler monkey (see below). On another occasion we assumed that an infant had died shortly after its mother had died and it was observed falling in a tree. We assumed one juvenile to have died having shown signs of physical weakness and struggling to keep up with the group. The remaining four deaths were all adults. We recovered the body of one female, which had several serious bite marks. Post-mortem examination by a veterinarian found the cause of death to be lung perforation, consistent, in terms of bite shape, breadth and depth, with an attack by another howler monkey (M Escorcia-Quintana, pers. comm., 2008). One adult male probably died after we observed it with severe open wounds resulting from an attack by two immigrating males. Another adult male showed signs of paralysis and lethargy before his assumed death. A further adult male showed signs of old age, lack of appetite and was unable to keep up with the group.

We registered deaths in most months of the year, but there was a clear peak between November and March when 75.3% of deaths occurred (Fig. 3D, Table 1). There were more deaths than expected during the period of energetic stress ($N = 14$), and fewer than expected in the rest of the year ($N = 4$) ($X^2 = 9.66$, $df = 1$, $p < 0.005$, $\varphi = 0.54$). However, the $r$ statistic for seasonality was not significant (unimodal $r = 0.08$, $p = 0.89$; bimodal $r = 0.25$, $p = 0.30$).

### Disappeared

We were unable to interpret the history of 27 individuals from the data, which we recorded as disappeared (Table 2).

## DISCUSSION

We present data on 11 years of demographic events in 10 groups of mantled howler monkeys living in an anthropogenic landscape in Mexico. Due to the discontinuous nature

of our sampling method, it is likely that we failed to record some events. For example, we might have missed short-term dispersal events, or births followed quickly by deaths. Also it is possible that some events were recorded incorrectly: e.g., when we did not observe an individual in a group for more than one month, we assumed that it had migrated, but it is also possible that it had died suddenly, or been killed by a predator or conspecific. However, given that the number of emigrations closely matched the number of immigrations, and that no natural predators of howler monkeys remain in Los Tuxtlas (*Cristóbal-Azkarate & Dunn, 2013*), we consider our assumption to be reasonable.

While acknowledging the limitations of our study, our data suggest a dynamic population with frequent demographic change, including a large number of migrations, births and deaths. While births were distributed throughout the year, they were highly seasonal, with a clear peak between October and December and a secondary peak in February. Another study carried out in a different area of Los Tuxtlas found similar results (*Carrera-Sánchez, Medel-Palacios & Rodríguez-Luna, 2003*). This suggests that the majority of conceptions that lead to births occur between April and June (Fig. 3A), coinciding with the annual peak in fruit availability and increase in ambient temperature (Figs. 2A and 2B). Accordingly, our data suggest that the higher energetic stress between November and March may be inhibiting the reproduction of females and that the improved conditions from April to June results in an increase in fertility. Other studies have also reported that the time of conception is associated with the availability of food and temperature in howler monkeys (*Kowalewski & Zunino, 2004*). This supports the idea that howler monkeys are income breeders (rather than capital breeders) and that they use energy acquired during the reproductive period for reproduction instead of stored energy (*Brockman & Van Schaik, 2005*; *Janson & Verdolin, 2005*). Similarly, the weaning of offspring would also occur in April and May, supporting the idea that the weaning of offspring in howler monkeys occurs at times of year in which the availability of high quality food is higher and the climate is more benign (*Kowalewski & Zunino, 2004*).

The mean birth rate of the study groups is within the range reported for other growing populations in the Neotropics (Table 3), which suggests that, in principle, our study population is not constrained by its reproductive output, and the IBI is also within the range previously reported for the species (Table 3). However, comparisons of birth rate and IBI across studies should be made with caution, owing to differences in methods. The death of an infant significantly reduced the IBI, a phenomenon also reported for other primate species (*Fedigan & Rose, 1995*).

We recorded numerous migration events, with both emigration and immigration being observed in almost all of the study groups. This included the groups that inhabited a forest fragment with no other groups, as they all received immigrants, and all but one were a source of emigrants. This suggests that, in our study landscape, howler monkeys are able to transfer between forest fragments. This behaviour has also been reported elsewhere for howler monkeys, and the probability of dispersal has been negatively related to the isolation distance of the fragment and positively related to the connectivity of the fragment and heterogeneity of the landscape (*Glander, 1992*; *Mandujano, Escobedo-Morales & Palacios-silva, 2004*; *Estrada et al., 2006*; *Mandujano et al., 2006*; *Asensio et al., 2009*). Accordingly,

we believe that the high levels of dispersal recorded in our study population are probably related to the high level of landscape connectivity.

The high number of migratory events that we observed is a good sign for the long-term viability of the population, as transfer among forest fragments may serve to mitigate the negative effects of forest fragmentation on howler monkeys, by improving access to resources and promoting outbreeding. Unfortunately, we were unable to determine the exact origin and destination of most migrations. Determining which groups and fragments are in migratory contact with each other, in addition to identifying important dispersal routes, would allow for better modeling of the dynamics of our study population and help identify priority areas for conservation. This gap in our knowledge should be addressed in the future with research focused on molecular genetic methods in addition to telemetry to follow the movement of individuals in the landscape.

Both natal and secondary emigration were common in our population. The fact that most juveniles leave their natal group is well described in the literature (*Glander, 1992*), but it was not until very recently that it was proposed that secondary dispersal may be a common and important component of the reproductive strategy of mantled howler monkeys (*Clarke & Glander, 2010*). The fact that almost half of all emigrations in our study population were secondary dispersals provides strong support for this hypothesis. While emigration was not found to be strongly seasonal, males and females showed clear peaks (males in January–February and August; females August–September) which preceded the peaks in immigration by less than two months (Figs. 3B and 3C), while the emigration of subadults and juveniles peaked in November, coinciding with the beginning of the period of fruit scarcity and higher levels of physiological stress (*Dunn et al., 2013*). This could suggest that the timing of adult emigration might be associated with factors determining the best time for transferring to a new group (e.g., resource availability and reproduction), while the emigration of subadults and juveniles might be driven by competition for food. However, we cannot rule out the possibility that the January–February peak in male emigration might also be associated with competition for food. Without more information on the life of solitary individuals in Los Tuxtlas, including data on the duration of this period for males and females, it is not possible to draw any definitive conclusions from these data.

Like emigration, immigration was not found to be strongly seasonal in statistical terms. However, for both sexes these events were more common during the primary and secondary peaks in fruit availability and consumption by howler monkeys in Los Tuxtlas (*Dunn, Cristóbal-Azkarate & Veà, 2010*), which suggests that resident individuals may be more willing to accept immigrants during periods of relative resource abundance. Moreover, the primary peak in male immigration (April–May) coincides with the time when most conceptions leading to births occurred. It is not clear whether in Los Tuxtlas immigrating males achieve alpha status immediately upon immigration as described in *Alouatta palliata palliata* in Costa Rica (*Glander, 1980*). However, several males were observed mating with females shortly after immigration (J Dunn, pers. comm., 2007) and, nonetheless, mantled howler males are not reported to monopolize reproduction (*Jones, 1995*; *Wang & Milton, 2003*). Therefore, the availability of fertile females may be driving, at least in part, the

timing of immigration of males. On the other hand, by joining the group several months before the onset of the period with the highest number of conceptions leading to births (April–May), the females may have more time to achieve an adequate position in the group to maximise their chances for successful reproduction when the environmental conditions are optimal.

The fact that we only recorded 3 immigrations by juveniles, but recorded 13 emigrations, suggests that the mortality of juveniles may be high during these periods, and/or that immigration into groups is easier for fully grown adults and juveniles may need to spend several years as solitary individuals before forming a new group or joining an established group (*Glander, 1992*). Although intense fighting has previously been reported between resident males and adult male immigrants (*Clarke & Glander, 2004*; *Dias et al., 2010*), and evidence from facial scarring and injuries suggests that fighting may be common in howler monkeys in Los Tuxtlas (*Cristóbal-Azkarate, Dias & Veà, 2004*), we only observed one such fight during our study, and apart from this occasion, we did not observe any males with injuries following an immigration event.

We registered 18 assumed deaths, which were predominantly infants, although we were unable to determine the cause of death for most of the cases. The fact that a group female was, seemingly, killed by a conspecific while carrying a 1-month-old infant is noteworthy, and may have been the result of an attempted infanticidal attack. However, this is speculative and, without more information of the context and details of the event, it is difficult to interpret. One male probably died after we observed it with severe open wounds resulting from an attack by two immigrating males. Ignoring these cases, which were seemingly the result of intraspecific aggression, deaths showed a clear pattern with 75% of total deaths, and 100% of adult deaths, occurring in the period of energetic stress. Thus, it seems that energetic constraints may be an important factor regulating the population dynamics of howler monkeys in the region.

Overall, our results suggest that the population of howler monkeys in Los Tuxtlas has increased during the eleven-year study period (though this increase is largely due to two groups). Moreover, we found migration events to be frequent between groups and fragments, despite the isolating effects of forest fragmentation. However, the study period was short relative to the long life span and slow life-history of howler monkeys, and the fragmentation history is relatively recent in the region, meaning that group size and composition may not yet be well suited to the current environmental conditions. Only studies covering several generation-times, which incorporate indices of health, reproduction and fitness (e.g., ecophysiology, molecular genetics) in conjunction with intensive data on demographic evolution, would allow us to fully examine the long-term conservation prospects of this population.

## ACKNOWLEDGEMENTS

This paper is dedicated to the wonderful memory of our colleague Dr Joaquim Vea-Baro, founder of this long-term research project, who died in February 2016. Joaquim was a passionate ethologist and primatologist, whose legacy in terms of both scientific output

and the training of students will continue for decades to come. We are grateful to all of the field assistants and researchers that, during their time in the field, have collected demographic data for our analyses: Lino Mendoza, Pedro Dias, Sira Vegas-Carillo, Blanca Hervier, Norberto Asensio, Iñaki Aldekoa. We also thank Ernesto Rodríguez-Luna and Rosamond Coates for their valuable assistance with our research and Carmen Galán for help with Fig. 1.

### Funding

The first census in 2000 was supported by the Zientzi Politikarako Zuzendaritza of the Basque Government (JCA). Subsequent studies that provided data for our analyses were funded by grants from the Spanish Ministry of Education and Science (PB98-1270, SEJ2005-01562), Fundación BBVA, The Isaac Newton Trust (JD), the Mexican Ministry of Foreign Affairs (CDB) and Barcelona Zoo (CDB). The funders had no role in study design, data collection and analysis, decision to publish, or preparation of the manuscript.

### Grant Disclosures

The following grant information was disclosed by the authors:
Zientzi Politikarako Zuzendaritza of the Basque Government (JCA).
Spanish Ministry of Education and Science: PB98-1270, SEJ2005-01562.
Fundación BBVA.
The Isaac Newton Trust (JD).
The Mexican Ministry of Foreign Affairs (CDB).
Barcelona Zoo (CDB).

### Competing Interests

The authors declare there are no competing interests.

### Author Contributions

- Jurgi Cristóbal Azkarate and Jacob C. Dunn conceived and designed the experiments, performed the experiments, analyzed the data, wrote the paper, prepared figures and/or tables, reviewed drafts of the paper.
- Cristina Domingo Balcells conceived and designed the experiments, performed the experiments, analyzed the data, wrote the paper, reviewed drafts of the paper.
- Joaquim Veà Baró conceived and designed the experiments, performed the experiments, contributed reagents/materials/analysis tools, wrote the paper, reviewed drafts of the paper.

### Animal Ethics

The following information was supplied relating to ethical approvals (i.e., approving body and any reference numbers):

This study is based on observational data and there was no direct interaction with the study subjects. We were verbally granted access to the study site by local communities,

landowners, and the Los Tuxtlas Biosphere Reserve, part of the National Commission of Natural Protected Areas of Mexico (CONANP). All research adhered to the American Society of Primatologists Principles for the Ethical Treatment of Non-Human Primates and to the legal requirements of Mexico.

### Field Study Permissions

The following information was supplied relating to field study approvals (i.e., approving body and any reference numbers):

We were granted access to the study site by local communities, landowners, and the Los Tuxtlas Biosphere Reserve, part of the National Commission of Natural Protected Areas of Mexico (CONANP).

### Data Availability

The raw data is included in the tables in the manuscript.

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
