# Peer review of "A demographic history of a population of howler monkeys (Alouatta palliata) living in a fragmented landscape in Mexico"

_PeerJ, doi:10.7717/peerj.3547_

## Round 0.1 · original submission · Major Revisions

Dear Dr. Azkarate,

We have now received two reviews of your manuscript. Both reviewers agree that the data set is valuable and of sufficient potential interest to warrant publication. However, both reviewers cite substantial revisions that will need to be made before the manuscript is ready for publication.

Overall, the reviewer's comments point to several major issues that you should attempt to address in a revision. I encourage you to carefully consider each of the reviewer’s points. In addition, I have a few additional comments of my own.

First, the motivation for the study should be more clear. You state several times that the species is endangered, but it is not clear how this status relates to the data you are collecting. I believe that if you sharpen up your description of the questions you are asking in the manuscript and the reasons you want answers for them, this will help not only in making the motivation clear, but in framing the results more conceptually. Why did you measure differences in birth rates among groups, what would it have told you if there were differences? Why did you measure differences in demographic rates across seasons? You say that knowing demographic rates is important for monitoring the conservation status of the species, but why, and how would the information be used? These are not difficult questions but you should not assume that the reader knows in advance the answers to them. Further, spelling things out will help organize and streamline your results, which both reviewers comments on a need for.

Related to this point about a conceptual framework (and to a comment by reviewer 1), you frequently mention the need for long-term data, but you never say why such data are important, how they should be used, or what ‘long-term’ means in this context. One possible point you’re making is that repeated sampling of the same population over time provides insight into population dynamics over time. However, you don’t look at any changes over time in demographic rates (although you do report that the population has grown over time), so this seems not to be your motivation for valuing long-term data. Another possible reason is that repeated sampling over time and over various ecological conditions provides more robust measures of demographic rates than one-time measures. I think this might be the point you are trying to make. However, it’s not clear, and I think it will strengthen the manuscript if you can try to identify what it is you think is important about ‘long-term’ data and talk about those features, rather than simply describing the data as long term.

You several times refer to the fragmentation of the habitat and draw conclusions about its effect on the population but it’s not obvious to me that these conclusions are warranted. Specifically, you conclude that the fragmentation has not affected the monkeys because the population has been the same size for the past 10 years. But this raises more questions than it answers. When did the fragmentation occur relative to your censuses? If the fragmentation occurred prior to 2000, how do you know whether the population changed as a result of fragmentation or not?

I also have a concern about your treatment of emigrations. You state in the discussion that “The fact that we only recorded 3 immigrations by juveniles, but recorded 13 emigrations, suggests that the mortality of juveniles may be high during these periods” – that is, you suspect that some of your emigrations are really deaths. Indeed, this also seems extremely likely to me. If an animal disappeared without signs of injury or illness you assumed it had emigrated – this does not seem like a conservation procedure and could strongly affect your results. This made it difficult to assess your results for emigration - I kept wondering how many of the apparent emigrations were deaths. Importantly, peak emigration rates occur at a time that overlaps with peak death rates.

Somewhat related to this, your conclusion that migration events are likely to be peaceable is confusing given that you identified a death that was associated with conspecific immigration.

Although the revisions that are requires may seem substantial, I believe that the valuable input of the reviewers will result in a much stronger paper if you can address their concerns.

Reviewer 1 ·

Basic reporting

The article needs substantial work in the writing.
The introduction does not provide enough background to be of interest to a broader field of knowledged.
The submission is self-contained, it is an appropriate unit of publication with interesting and abundant information.

Experimental design

The article describes primary research with a clearly defined question
Methods are very adequate, they need work in presentation and some work on distinguishing statistical results from actual data.

Validity of the findings

Don’t know if the data are being made available in a discipline-specific repository.
The conclusions need some work, some more connection between data and fragmentation implications is needed.
Because the journal accepts “negative” results, I encourage the authors to even more distinguish statistical probability from actual data.

Additional comments

There is significant work to do in writing style.
This is data that merits to be known by the primatology community, but I am not sure if the manuscript is of interest to a wide audience as I suspect is the audience of PeerJ.
I suggest reorganization some of the result sections. May be having a section on seasonality where the various demographics are considered. As it is it becomes a bit repetitive.
I know it is common for people to say "a total of xx deaths", but never understood why "a total" is necessary. Do people ever record "partial" births or deaths? Why not "we recorded 78 deaths"?

INTRODUCTION
I would like to encourage the authors to avoid using "long-term" and instead consider something that is either better defined or more related to the species under consideration. The term "long-term" continues to generate a lot of confusion because it is undefined and irrelevant if not considered in view of the lifespan of the species considered.

l.33 "several generations..." may be reference to one or two examples?
l37-39 there have been studies of A. caraya in Argentina spanning since the early 80's
l43-46, rephrase, may be two sentences? each one shorter?
l47. I am not familiar with the journal policy on this but I firmly disagree with the idea that us, the primatologists, the ones who presumable know about primate population biology, will build a rationale about the urgency to protect a population, species or subspecies based on an IUCN listing. The IUCN listings are seldom informed by solid science and when they are, it is science written by our colleagues. If there are solid scientific data to argue that this subspecies is critically endangered, then the authors need to cite the authors who published those data. If there are no solid evidence (something quite common for IUCN listings) then the argument is not valid for a scientific journal.
l57-58 here the authors are implicitly agreeing with me. If long-term demographic data is "crucial" for understanding the conservation status of the subspecies because we do not have those data yet, how can the assessment of critically endangered by IUCN be trusted? I urge the authors to let the IUCN listings be used by policy makers and the media, and us focus on the real science that needs to be done.

METHODS
l69 Universidad Veracruzana?
l72. How man subspecies have been described in the peer-reviewed scientific literature? That is the reference the authors need to use.
Some of the information on the study species should be moved to the introduction where they are already talking about this.
l80 which primates? Alouatta spp. or the subspecies? The sentences refer to the species, not the subspecies.
l87 which howler species? The text mixes discussion of the subspecies under consideration with references to many different Alouatta taxa.
l108 delete "is"
l155 Information provided in Fig. 2b needs to be presented before the reader is pointed at the figure.
l117 not sure journal's guidelines, but isn't it et al for more than 2 authors?
The methods section also need work on writing style.
l134-135 methods need to be made full available, in the text or supplementary information, or other references, but not "upon request"
l153-154 rephrase, not clear
l193 spell out percentage
l196, remove double parentheses, use commas instead, check across manuscript
l217 again I am not familiar with the journals' position on statitiscal reporting, but I urge the authors to review the abundant literature on the many problems of having an arbitrary value of 0.05 determine whether things are significant or not. Results of statistical analyses should include full consideration of the variation in the data (C.I., complete descriptive stats, effect sizes, etc.). For the specifics of the questions being explored here it simply does not seem the best approach to set up a test of whether there is or there is not a relationship with seasonality. There can be no doubt that the reproductive biology of a primate will be somewhat related to changes in climate and food, the question of interest is not is there a relationship or not, but what is the strength of that relationship.

RESULTS
Table 1 is a good summary of data.
When differences are discussed, distinguish whether you refer to statistical differences or biological ones. In l.233 for example, when discussing birth rates, this is obviously important. I will argue that a difference of rates from 0.16 to 0.58 has to be biologically important. It means that one group is producing on average almost four times more infants that then other. The data need to be considered some more (like the tables do, with good descriptive stats), not just summarize the results with a NHST result.

l234. use decimals critically, in text and tables. provide those decimals that give useful information or are justified given the methods. An IBI of 21.61 is neither, your methods as explained did not give you the power to estimate the IBI to the second decimal.
for the statistical tests, I encourage the authors to provide Confidence intervals and effect sizes. Doing that will point at the fact that the "non-significance" of differences in birth rates and the "signficance" of IBIs is most likely almost exclusively related to differences in sample size.

Table 2, not sure you should provide all the data from other studies.

Fig. 3, I wonder if you should provide averages per group for the seasonality. With the variation in birth rates there is, you may want to examine seasonality in each group first, only then plot averages across groups with s.d.

l,.242-243, but it is not the test that suggests a bimodal distribution. You have just explained in the sentence before how the distribution is bimodal. The tests, and the associated p-values, are only adding a probability statement. Again, I suggest focusing and discussing the data first, then providing the stats with good information on the uncertainty associated with the probability value.
l243-244 this goes in the discussion, not results.

l246 Provide the data, the results, not only a statistical outcome.

Emigration, may be it can all be summarized in a table.
l261-262, methods not results
l264 the paragraph on seasonality of emigrations needs to be rewritten, bringing first to the author the actual results, the supported by stats when and if necessary.

l276 "in all forest fragments". Also check punctuation, there is need for commas in many places. There is quite some work to do in revising the writing style.

Immigration
the paragraph mixes results and methods.

l297-99, needs rephrasing, punctuation.

l314-315, methods.

DISCUSSION
Incorporate first one-sentence paragraph with other paragraphs.
l322-26, this has been already said in the results.

l356. are the animals migrating from other fragments, or can they be from fragments that do not exist anymore? Your fragments acting as a sink?

l379. My looking at the data made me think of strong seasonality. I wonder why the authors consider this not to be strong seasonality. There is variation, of course, but there is also room for variation due to the sampling methodology and the "estimation" of the dates that get analyzed. Again, I encourage them to consider the extent to which the patterns observed are profoundly seasonal in biological terms.

l415, but you have almost 30% more individuals that at the beginning? from 68 to 92? why is this not an increase? 5 groups increased, 4 stayed the same, only 1 decrease. Have you looked at birth rates over time? That may be a good indication of growth through recruitment.

l418 Yes, agree! that is why this "long-term" reference is useless and we should all stop using it.
l420 change to "only studies covering several generation-times" or something like that...

·

Basic reporting

This is a well-written paper containing original information on one of the few long-term demographic studies on howler monkeys. It reports data on 10 groups followed during 10 years in Los Tuxtlas, Mexico, evaluating the status of the mantled howler population in a fragmented habitat and the influence of seasonality on demographic parameters. The information on howler demography contained in the article is highly valuable; the manuscript presents several interesting points of discussion and represents advancement in our current understanding on howlers’ population ecology. Thus, overall the paper could be an important contribution in the field of primate studies, and should be accepted for publication. However, the current version may be improved should the authors accept to make some minor changes and address some points suggested in the comments below.

Experimental design

Authors should mention the study site is located in Mexico in the abstract, somewhere.
The context of fragmentation is very important, since apparently authors want to assess whether their howler study population is growing, stable or decreasing in a situation of highly modified habitat. Nevertheless, the manuscript lacks of a fair description of the fragmentation scenario where their study has been carried out. 1) How recent is this process? 2) What happened during the 10 years-period covered by their research, was the habitat modification already stopped, or was it continually reducing howlers’ habitat? 3) On what basis did the authors (or their colleagues) select the fragments with howlers? I guess there are some fragments in between that also contain howlers and were not surveyed, but it is not specified. 4) It may be worthwhile introducing some index or measure of connectivity, since sentences like “Nevertheless, compared to many other fragmented landscapes, it retains a relatively high level of connectivity” (lns 103-104) do not allow the reader to understand the real degree of disturbance from a howler perspective.

Study sites and Data collection. Authors say that they studied 10 groups during 10 years, but then mentioned that they started collecting data in 2010 and continued until 2011; this would make a total of 11 years. Also, it is not clear what is stated in lns 128-130: what does it mean “whenever possible”? It should be more rigorously described the collection data protocol in the Methods section. Afterwards, it is shown that there was quite a large variation in the sampling effort among the 10 groups, in terms of how long each of them has been followed and with which intensity. To my opinion, this could have introduced a bias in the evaluation of several of the demographic parameters estimated. Even though the uneven sampling effort is taken into account in seasonal comparison by weighing the original values according to different frequencies of visits, some parameters such as IBI may be overestimated in some groups compared to others (this could happen when comparing among groups, Ln 195).

Lns 139-143. Based on my own experience, it is hardly possible to identify at the individual level so many individuals (up to 92) of howler monkeys. Primates of this genus are generally quite difficult to distinguish, unless they have very peculiar features, such as scars, marks or pelage color. In fact, in many long-term studies on howlers, where researchers survey several groups on a regular basis to gain data on demography, individuals are marked for identification purposes (e.g. Glander 1992; Kowalewski 2007). Also, in a previous paper by Cristóbal-Azkarate et al. 2007, the authors acknowledge that there were some females that could not have been identified at the individual level (table I). What about the 10 groups followed in this study? Could the observers be 100% confident of accurate identification of all group members? Also, how many observers were involved in these monthly surveys? Any measure of inter-observer reliability for individual identification and age estimates?
It would be nice to add a few lines to further detail how were monthly visits conducted (one day/visit? How many hours spent?) and whether there was a previous habituation period for each of the followed groups.

Lns 145-184. It is logical that authors had to make assumptions, given the unbalanced sampling design among groups, and the discontinuity in their data collection. However, some of the assumptions they made can be questionable.
Ln 155-158. How did the authors control for discontinuity in data recording for some groups when they calculated annual IBI per female?
Lns 161-164. In a lapse of 1 month, an individual could have got injured or ill. It is difficult to discard these possibilities in this time interval. But above all, is there any chance individuals are predated? I know that in the region of Los Tuxtlas, the main top predators have disappeared (jaguars, pumas and harpy eagles), however in Cristóbal-Azkarate & Dunn (2013) page 84, it is reported that some events of predation may take place when monkeys travel on the ground and get caught by coyotes or tayras. Especially in groups occupying small fragments with a significant edge effect, could it be possible that occasionally monkeys coming low to the ground are predated?
Lns 176-180. Similarly, except in cases 1) and 2), it is not completely straightforward to assume an individual injured or ill is necessarily dead if disappeared. Sometimes, it is surprising the recovery potential of severely injured individuals among wild primates. It may be perfectly plausible that, especially after a fight, an injured individual emigrate to another group, probably after staying solitary for a while.
Finally, any chance that some births went undetected? Death of infants within the first month of life is not uncommon in wild primates, and it could probably bias the estimate of IBI and births events.
I am not saying assumptions cannot be made, but limitations should be acknowledged by the authors.

Validity of the findings

Lns. 373-376. This seems highly speculative. Any explanation of these differences between adults vs subadults/juveniles? Different pressures? These speculations should be more justified from a biological point of view. Otherwise, there is no point in speculating farther.
Finally, although the study has been carried out in a small number of fragments, do they authors have any analysis on the influence of fragment size on all demographic parameters?

Much of the interpretation of the results concerning the variation in demographic parameters refer to food resources availability, as a potential predictive factor influencing emigration, immigration, birth, etc. Since data on resource availability as well as temperature and rainfall are available, it could be worthwhile examining directly these relationships (using correlations or regression analyses?).

Additional comments

No comments.

---

## Round 0.2 · Minor Revisions

One reviewer provided a number of comments and suggestions for revisions, and I draw your attention to these.

1. The reviewer comments that the Introduction perhaps overstates the relevance and importance of your data. I see this point, but feel that this problem is offset by the fact that you now have produced a streamlined, clearly written justification and context for the study. I leave it to your discretion to consider the reviewer’s comments and point of view, and decide how to modify the introduction in response.
2. The reviewer requests that you present more data, and does in the body of the manuscript itself rather than as a supplemental table. I am sympathetic to this request; your Supplementary Table is not large, and may not be even when you accommodate the reviewer’s request to provide age/sex data on the demographic events.
3. The reviewer comments ‘I still don’t think the data on other studies is necessary. We must understand that comparisons with other studies should begin by ruling out methodological considerations for our findings first. And the authors are not considering the breadth of methods into that list of studies.’ I appreciate this point but also undersetand that you are trying to contextualize your study. Perhaps the reviewer’s point can be incorporated into your text to make clear that you understand that study-to-study differences are likely to result from methodological differences.
4. The reviewer requests whether there is any scientific literature supporting the critically endangered status of your study species; if so, please cite it.
5. The reviewer also makes specific comments on particular sentences in the manuscript and suggests that you proof read carefully. Please include responses to these in your revision.

Reviewer 1 ·

Basic reporting

The article still needs work in the writing.
The introduction still reads a bit like “we need more data because it is going to help conservation”
I still think that the manuscript contains interesting data worth publishing, ideally without so many claims that sometimes feel a bit pushed to fit the hypothesis of energetic stress that one of the coauthors has published before.

Experimental design

The article describes primary research with a clearly defined question
They still need some work on distinguishing statistical results from actual data.

Validity of the findings

I appreciate the opening paragraph of the discussion pointing at limitations.

Additional comments

I have now read the revised version of the manuscript submitted by Cristóbal-Azkarate and colleagues, presenting demographic information on 8 groups of mantled howlers monitored during 11 years. The strength of the article lies on the data, it is weaker on the attempts to link presumed demographic changes to seasonal factors for which there are no new data, but data quoted from other studies.

This data set has the potential to be used for many years to come by anyone interested in Alouatta or any other primate taxa. It is for that reason that I urge the authors to present the data more clearly, more openly and the raw data. Yes, there are limitations, every study has them. But it is a very interesting data set.

I think that it would be easier for the reader to have access to the actual data as opposed to “weighted data”. Why not show the number of births per group per month, it is only 49 data points? I don’t understand why and how the number of visits to groups, assuming these are well monitored groups, can have a profound influence on estimating birth rates. Females are pregnant for many months, therefore detectable and then infants can be recorded as births as well for a long time, in fact that is what they have done. Then just show the births clearly in the manuscript, not in the Supplemental table in a way that is difficult to relate to the claims being made.

A similar point to present to the readers the findings on how the different age classes and sexes differed. The data are not clearly summarized, TS1 is not clear. Summaries per group are important, summaries per age, per sex.


TS1 I downloaded it but did not find the legend, I assume “dark grey” means monitored in that month? I think the authors should produce a table that shows numbers for the demographic events being considered. A more clearly organized TS1 should be part of the manuscript.

Many of the results are presented in terms of age classes and sexes, but the data are not available. TS1 does not provide distinction between ages or sexes. It is a shame, the authors need to share the data more clearly, openly, let the reader go and see who dispersed at what age, what sex from where. This is not a huge data that cannot be shared in that way.

I still don’t think the data on other studies is necessary. We must understand that comparisons with other studies should begin by ruling out methodological considerations for our findings first. And the authors are not considering the breadth of methods into that list of studies.

If there are no data on many groups to evaluate seasonality per group, then why not limit the analyses to those groups that provide solid data? I don’t agree one can say that “seasonal factors affect all groups similarly” when the study is built around the premise of finding out if seasonal factors affect the groups…….

Although in some places there is now some consideration of data first, there are many where there is not. The authors continue to have, in my opinion, too much reliance on “statistical significance”. On one hand they acknowledge limitations, but then pool all the data together and run NHST which is extremely influenced by sample size.

The manuscript could have benefitted from a careful, thorough proof-reading. I am indicating some editorial typos below, but I stopped recording them. The authors ought to check that.

L28, the word “significant” is loaded with statistical implications, may be “important”?

L57-58, the authors talk about “effects” but they acknowledge in one of their comments that they cannot do this (“We do not have a large enough sample, with enough reliable replicates, to be able to carry out a robust study of the “effect of forest loss/fragmentation” on howler monkey demographics (i.e., using demographic variables as dependent variables in models with habitat characteristics as independent variables).” I agree they should not be talking about effects which imply a potential causality they cannot evaluate, but merely some possible association or relationships.

L63-64. I disagree with the use of IUCN criteria for developing a SCIENTIFIC argument, it is not for better or worse, it is for worse. It is not based on solid science, but on policy and fund raising needs of a small group of people in one single organization that usually organizes and fully funds the workshops for making the classifications. The editor will decide what they want the journal to do. If there is no disagreement in the scientific community why not cite those sources?

L64-66. I disagree. Given that the existing data cannot allow us to assess the effects of fragmentation, how will the data allow us to assess the ability of these primates to adapt.

L67 replace “effect” with association or relationship.

L70. Are there no sources of information from colleagues publishing on Alouatta to send the reader to read more about the distribution of the genus?

L109 replace “significant” for important or something else.

L135-36 “in the region” twice…

L151 check formatting, eg. Outside parenthesis

L166 “we assigned it an age and sex”

In order to control for the effect that our slightly unbalanced sampling effort could have on the seasonality data, we weighted the original data by dividing the frequency of events per month by the number of different visits to the same group within a month (mean ± SD average visits per month = 37.6 ± 3.1, range = 35 – 42; Table S1). We used these weighted values to calculate the percentage of demographic events in each month.

L214 “we considered only those events….”, make a table clearly showing which events are being considered for the analyses. The data need to be more readily available for anyone to redo the analyses.

L223-224, “can be, as a rule of thumb, considered as small….etc”

L232 “statistically significant differences”

L244 “statistical significance”

L252, I guess it depends on the journal’s policy, the literature stating again that arbitrary cutoffs are meaningless, damaging and should not be used is growing exponentially. The most recent, hopefully game-changing statement has been made by nothing less than the American Statistical Association stating that:

L270-271, another reason for expanding our consideration of the biology behind our data is that if 10 years of data on 8 groups….

L277 “and found”. Why not phrase in a manner that is easier to see the relevance? There is a one-year difference in the pace of reproduction, that is most likely important and it seems statistically significant as well.

L312, emigrations peaked in August because there were 11 of them MT1, shouldn’t that absolutely extreme outlier be explained, considered?

L318-319, please edit to share with the reader the biological relevance of the results.
L342 “a one-month old”
L357, I suggest removing from the results reference to the “period of energetic stress” since this manuscript provides no data about that. I can see connecting previous published work with this in the discussion, but I have to say that I started feeling that the authors were sold on the idea that the seasonality matches the “energetic stress” and all analyses and presentation of data are structured around that, instead of actually examining critically the data to see if it matches. Which takes me back to one of my main comments, I would like to see the data, the raw data so that I can evaluate the results on seasonality better.


Figure 2, axis needs to say is mean values. The legend does not describe well the graph, it does not mention min and max. For 2b the legend and the Y axis do not match

Figure 3
L728 “they”

Table 2, why two decimals for the mean birth rate? Also the table talks about “birth per female per year”, but I assume it is not that but a weighted estimated. They need to be consistent, these are the data that then end up in a table in a book and people don’t know that these are not raw straight birth rates, but weighted ones.

The IBI 95% intervals need checking, I don’t understand how they can be so broad with two data points for example.

·

Basic reporting

In this second revised version of the manuscript “A demographic history of a population of howler monkeys (Alouatta palliata) living in a fragmented landscape in Mexico”, the authors have done a noteworthy effort in addressing every single issue raised by the reviewers and the editor. I find the manuscript to have improved a lot compared to the first version. Especially, it has benefit from the thorough work done in better defining the conceptual framework of this study, the importance of such long-term data, and the potential use they have for conservation status evaluation. I also appreciate the acknowledgment of all the possible limitations of their data and results, due mainly to the fact that the study was not explicitly designed to answer their questions. Overall, I find the manuscript to be suitable for publication in PeerJ in its current form. Congratulations to the authors for the excellent job done in critically applying all the suggested changes or exhaustively explaining the reasons for not following reviewers’ indications when they considered not adequate.

Experimental design

The authors have responded to one of my concerns about the description of the fragmentation context of this study, as well as comments about other aspects of possible limitations of demographic parameters estimation.

Validity of the findings

Thanks to the reorganization of the Introduction and the Discussion, authors have explicitly addressed why their results are important and what are the future challenges to improve and increase our knowledge about A. palliata mexicana's population biology and conservation status.

---

## Round 0.3 · accepted · Accept

Dear Jurgi and Jacob, I am satisfied with the revisions you have made to the manuscript and am pleased that you've completed it. I know it was important for you to complete this in honor of the 11 years of work on this population.